# Inside-out: Antibody-binding reveals potential folding hinge-points within the SARS-CoV-2 replication co-factor nsp9

Yue Pan[2], Indu R. Chandrashekaran[2], Luke Tennant[1], Jamie Rossjohn[1,3], Dene R. Littler[1]*

1 Infection and Immunity Program & Department of Biochemistry and Molecular Biology, Biomedicine Discovery Institute, Monash University, Clayton, Victoria, Australia, 2 Monash Institute of Pharmaceutical Sciences, Monash University, Parkville, Victoria, Australia, 3 Institute of Infection and Immunity, Cardiff University School of Medicine, Heath Park, Cardiff, United Kingdom

* dene.littler@monash.edu

**Data Availability Statement:** All files are available from the Protein Data Bank with accession number 8DQU.

## Abstract

Nsp9 is a conserved accessory component of the coronaviral replication and transcription complex. It is the predominant substrate of nsp12's nucleotidylation activity while also serving to recruit proteins required for viral 5'-capping. Anti-nsp9 specific nanobodies have been isolated previously. We confirm that their binding mode is centred upon Trp-53 within SARS-CoV-2 nsp9. Antibody binding at this site surprisingly results in large-scale changes to the overall topology of this coronaviral unique fold. We further characterise the antibody-induced structural dynamism within nsp9, identifying a number of potentially flexible regions. A large expansion of the cavity between the s2-s3 and s4-s5 loops is particularly noteworthy. As is the potential for large-scale movements in the C-terminal GxxxG helix.

## Introduction

Throughout 2020–21 the global spread of SARS-CoV-2 inundated health care systems worldwide [1, 2]. Fortuitously the vaccines developed against the Spike protein's receptor-binding domain have proven highly effective, significantly reducing morbidity and mortality from COVID-19 disease [3–5]. Increased study of coronaviral biology has led to better understanding of this virus alongside rapid development of new therapeutics that may help defend against emergence of novel viral strains [6].

The coronaviral proteome consists of structural proteins that make up the virion and non-structural proteins (nsp) that are only produced inside host cells. The former facilitate viral egress, dispersal and infection, the latter viral transcription, replication and immune evasion. When a virion infects a new host cell the ssRNA coronaviral genome is first transcribed by host ribosomes producing the non-structural proteins as a self-cleaving PP1ab polyprotein. The essential viral RNA-dependent polymerase, contained within nsp12, is separated out of this polyprotein and together with nsp7 and nsp8 becomes the core components of the replication transcription complex (RTC) [7]. A molecular machine that duplicates genomic mRNA

**Funding:** The author(s) received no specific funding for this work.

**Competing interests:** The authors have declared that no competing interests exist.

and produces subgenomic transcripts. Alongside its nsp7,8,12 core a cast of other viral proteins associate with the RTC to add further functionality [8–11]. For example, two copies of nsp13 helicase stably associate [12] aiding template unwinding and proofreading.

Coronaviral nsp12 consists of two domains, an RNA-polymerase domain and an N-terminal pseudokinase *Nidovirales* RdRp associated nucleotidyl transferase domain (NiRAN) [7, 13]. Nsp9 is a small accessory factor that associates with the latter [14]. Nsp9 has a unique viral fold and was originally proposed to be an RNA-binding protein [15–18], more recent work suggests it is a central element to viral mRNA cap-formation [10, 14], making it a potentially viable therapeutic target [18, 19]. The SARS-CoV nsp9 homologue (nsp9$_{SARS}$) is essential for viral replication [9, 14, 20], with perturbation of a conserved protein-protein interaction motif (GxxxG) reducing viral titres in replication assays of both SARS-CoV [20] and SARS-CoV-2 [14].

The means by which SARS-CoV-2 forms its $^{7ME}$GpppA$_{2'-O-ME}$-RNA cap has only recently been reconstituted *in vitro* using recombinant proteins [21]. Nsp9 appears to be the primary substrate of nsp12's NiRAN domain [14] binding via its sole C-terminal helix allowing nsp9's N-terminal residue to insert deep into the catalytic site and act as an acceptor [9, 10]. The NiRAN pseudokinase has diverged from a standard kinase fold, the domain still binds nucleotides and has been described as having three related but distinct catalytic activities [9, 14, 21].

The current model for NiRAN mediated 5'-cap formation has nsp9's amino terminal Asn-1 residue reacting with a 5'- triphosphate end of viral mRNA to form a covalent adduct releasing PP$_i$ [9, 21]. This intermediary nsp9-pRNA is resolved by a second NiRAN catalyzed reaction cycle whereby a GDP molecule attacks the adduct's high-energy P-N phosphoramidite bond releasing nsp9 and forming the core GpppA-RNA 5'-cap structure. A similar capping mechanism occurs in rhabdoviruses using enzyme-RNA intermediates and is termed guanosine 5'-triphosphatase and RNA:GDP polyribonucleotidyltransferase (PRNTase) activity [22]. In this SARS-CoV-2 capping model the NiRAN domain and nsp9 act together as a PRNTase with catalytic residues and adduct accepting residues residing on different protein chains [21]. Further methylation reactions are subsequently required to form the functional 5'-cap, which are catalyzed by nsp14 and nsp16 methyltransferases [21]. The 3$^{rd}$ NiRAN catalyzed reaction is an observed Mg$^{2+}$ or Mn$^{2+}$-dependent NMPylase activity whereby nucleotide monophosphate are added to Asn- again releasing PP$_i$, the molecular function of this activity is less clear but may involve polyA addition [14, 23].

5'-mRNA cap formation is vital for viral replication and nsp9's unusual and integral role in this process renders it of therapeutic interest. Nanobodies are isolated variable heavy domains from camelid immunoglobulins, also termed VHHs. Several nanobodies have previously been investigated for their potential to inactivate coronaviral proteins [24]. As reagents Nanobodies can be highly specific and nsp9-reactive reagents may aid understanding of its PRNTase role and could represent starting points for broad antivirals [19]. A cohort of anti-nsp9 specific llama antibodies had previously been derived following challenge with recombinant sulfhydryl-free nsp9 [25]. Further NMR-based characterisation of these nanobodies highlighted a number of potential epitopes with Trp-53 being central to recognition [25].

Herein we present the crystal structure of one of these nanobodies in complex with the SARS-CoV-2 nsp9 (we term nsp9$_{COV19}$) and confirm that Trp-53 is a major feature of an extensive antibody-binding interface in which the CDR3-loop forms an extended β-sheet interaction. Surprisingly, nanobody binding induced large-scale topological changes to nsp9 within its unique coronaviral fold. This process distorts all NiRAN-interacting elements of nsp9 [9]. It is not presently clear if this is induced artificially by VHH-binding or whether it is a trapped alternate structural state, or folding intermediate. The potential flexibility of this essential and dynamic coronaviral protein is described and discussed herein.

## Results

### Structure of the nsp9 VHH$_{2nsp23}$ complex

A number of anti-nsp9 antibodies have been reported with NMR-based studies suggesting potential epitopes [25]. To better characterise the nanobodies binding mode, we recombinantly expressed and purified nsp9 and anti-nsp9 VHH$_{2nsp23}$. The antibody and antigen were co-complexed on gel-filtration then crystalized. X-ray diffraction at the Australian synchrotron MX2 beamline allowed data to be obtained at 2.4Å resolution (see S1 Table and Experimental procedures for details).

The crystals diffracted in space group P6 with phases obtained using a llama VHH model lacking CDR-loops. Clear electron density was observed for the missing loop regions allowing them to be built and refined accordingly. Both components where present in our crystals when run on SDS-PAGE (Fig 1A) but our initial attempts to obtain the antibody-bound nsp9$_{COV19}$ structure was unsuccessful. Despite this, clear electron density was observed next to the $^{VHH}$CDR3 loop for two β-strands and the nearby major nsp9 α-helix. Subsequent rounds of building and refinement allowed a model of the VHH$_{2nsp23}$-bound form of nsp9$_{COV19}$ to be built and refined.

The final model has two copies of nsp9$_{COV19}$:VHH$_{2NSP23}$ complex present within the asymmetric unit (Fig 1A). The quality of the electron density facilitated placement of all VHH residues, however the P6 space group creates a crystal packing with a 90Å diameter solvent-filled channel. Although most of nsp9$_{COV19}$ could be easily placed, structural elements immediately adjacent to the large solvent cavity displayed a degree of electron density smearing. The precise position of residues 1–20 and 76–87 is not currently modelled although its approximate position is observable. The two copies of the VHH$_{2nsp23}$:nsp9 complex within the asymmetric unit overlay but have differences in some sidechains positions.

### Epitope mapping of nsp9 VHH binding sites

The binding site of VHH$_{2nsp23}$ upon nsp9$_{COV19}$ is extensive resulting in a buried surface area of 1380Å$^2$. Residues within both the $^{VHH}$CDR2 and $^{VHH}$CDR3 loops make up the majority of the binding interface (Fig 1B, 27% and 66% respectively). VHH$_{2nsp23}$binding was seen to occur via an extended antiparallel β-sheet interaction that aligns the $^{VHH}$CDR3 loop with the s4 and s5 nsp9 strands (Fig 1A). Residues $^{103}$YYFST$^{107}$ of $^{VHH}$CDR3 are antiparallel with the nsp9$_{COV19}$ s5-strand and form four β-sheet H-bonds and a backbone-sidechain interaction with $^{VHH}$Thr-107 (Fig 1C). Specificity is provided by further sidechain interactions between the three $^{VHH}$CDR3 aromatics and the N-terminus of the αC helix. Here $^{VHH}$Phe-101, $^{VHH}$Tyr-103 and $^{VHH}$Phe-104 all form van der Waals contacts with conserved nsp9 residues Leu-97, Lys-92 and Gly-93 respectively (Fig 1C).

The $^{VHH}$CDR2 loop of VHH$_{2nsp23}$ also contributes significant contacts, but unlike the $^{VHH}$CDR3 loop these are entirely sidechain mediated. Trp-53 of nsp9$_{COV19}$ is clamped between the hydrophobic residues $^{VHH}$Met-50 and $^{VHH}$Ile-52 (Fig 1C). The aliphatic Cβ portions of $^{VHH}$Ser-57 and $^{VHH}$Asp-59 also facilitate binding of the indole ring confirming this residues contributions to the VHH$_{2nsp23}$ epitope (25). Further specificity for nsp9 is provided by H-bonds between the carboxyl group of $^{VHH}$Asp-59 and the hydroxyl of Tyr-66; and between the carboxyl group of Glu-68 and the hydroxyls of $^{VHH}$Thr-33 and $^{VHH}$Ser-53 (Fig 1C). Together at least 8 epitope specific H-bonds, 1 salt bridge and numerous van der Waals contacts contribute towards the anti-nsp9$_{COV19}$ specificity of VHH$_{2nsp23}$. All binding-epitope residues are maintained in nsp9$_{SARS}$; while residues homologous to Trp-53 and Glu-68 exist in nsp9$_{MERS}$, other residues making up the s5-epitope are less conserved (S2 Fig).

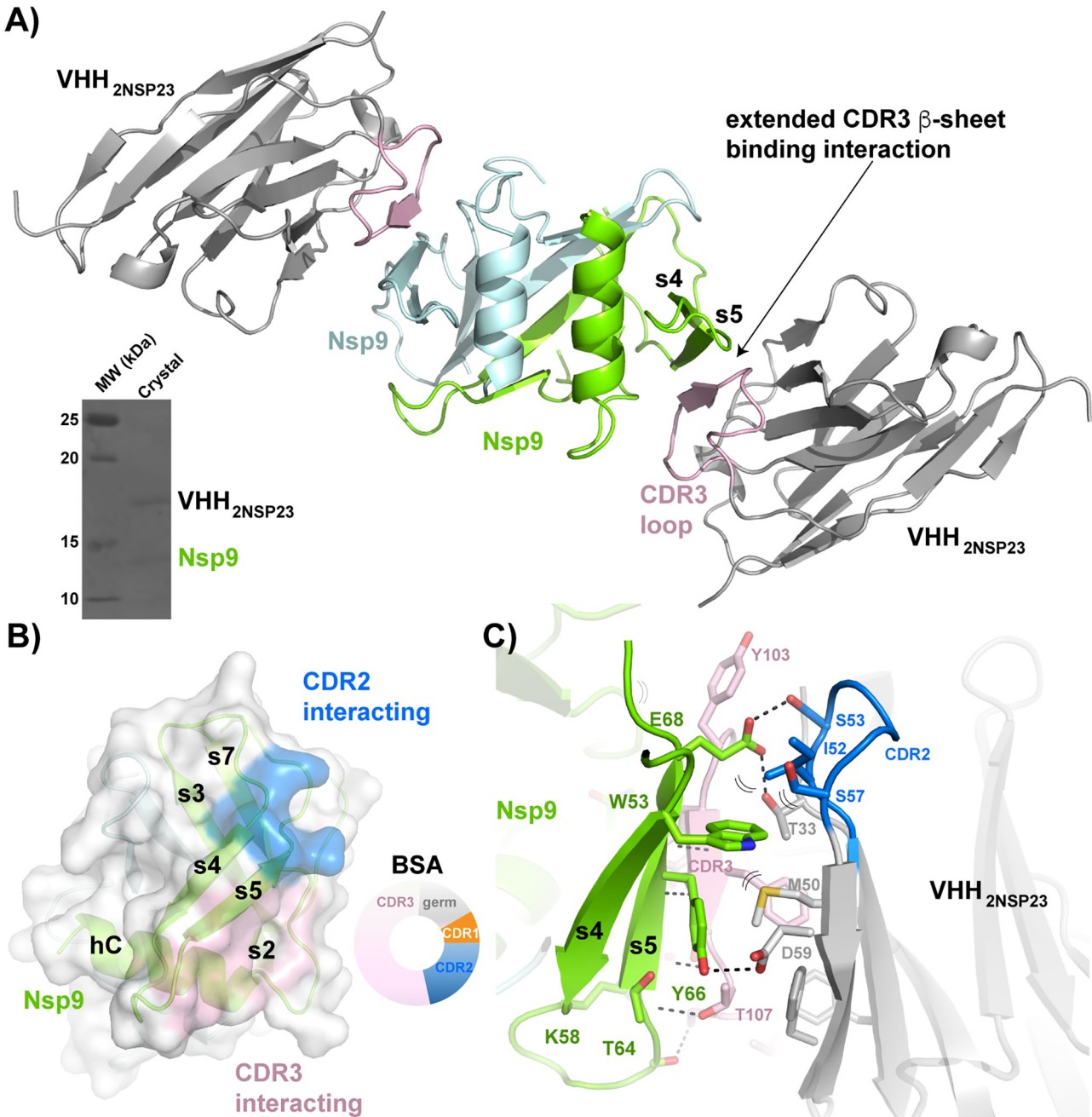

**Fig 1. VHH$_{2nsp23}$-bound SARS-CoV-2 nsp9$_{cov19}$. A)** Cartoon representation of the two VHH$_{2nsp23}$: nsp9$_{COV19}$ complexes within the asymmetric unit of the crystals. The two VHH$_{2nsp23}$ subunits are shown in *grey* with the extending $^{VHH}$CDR3β-sheet interacting residues in *pink*. The two nsp9$_{COV19}$ subunits are coloured *green* and *light blue*. The insert shows the intact complex run on SDS-PAGE after data collection. **B)** An overlayed surface representation of nsp9$_{COV19}$ coloured according to the regions bound by the VHH—$^{VHH}$CDR2 interacting residues are highlighted with *blue* colouring and the $^{VHH}$CDR3 residues with *pink* colouring. The doughnut chart insert shows the respective buried surface area percentages. **C)** An enhanced view of the residue interactions across the VHH$_{2nsp23}$: nsp9$_{COV19}$ interface. Hydrogen bonds are denoted with dashed lines and van der Waals interactions with double brackets.

## Changes to the nsp9 fold following VHH binding

The native unliganded form of nsp9$_{COV19}$ consists of a mini β-barrel in which the s1 and s6 strands wrap around the sole C-terminal α-helix (Fig 2A). This form undergoes structural

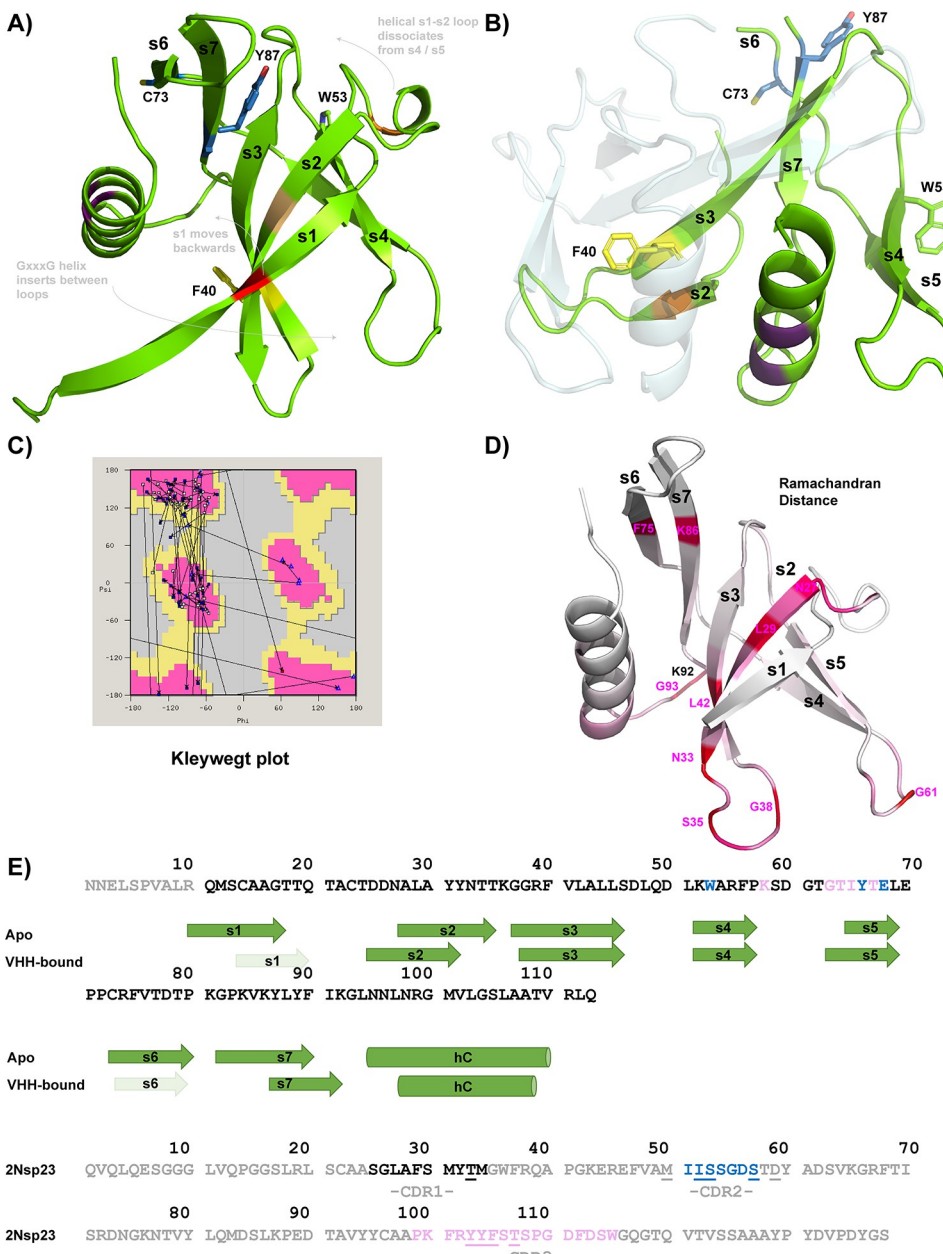

**Fig 2. Architectural changes to the nsp9<sub>cov19</sub> fold within the VHH<sub>2nsp23</sub>-bound complex. A)** Cartoon representation of the canonical coronaviral nsp9 fold with structural elements labelled. The position of key residues are noted with coloured highlights. **B)** An equivalent representation of a subunit of the VHH-bound nsp9<sub>cov19</sub> structure is displayed showing the opening of the mini β-barrel and insertion of the αC helix between the s2-s3 and s4-s5 loops. **C)** Kleywegt plot highlighting the extent to which residues undergo movements in Ramachandran space. **D)** The Ramachandran distances between the two states are mapped onto the canonical nsp9<sub>cov19</sub> structure using a *white->pink->red* gradient where the upper end of the scale is theoretical maximum. The identity of hinge residues are labelled. **E)** *Top*: The extent of secondary structural elements within each state of nsp9<sub>cov19</sub> are displayed under its sequence. *Bottom*: the sequence of 2nsp23 is shown with CDR loops coloured and labelled. VHH<sub>2nsp23</sub> residues interacting with nsp9 are underlined and their corresponding interaction partners coloured according to CDR interface.

rearrangement in the VHH$_{2nsp23}$-bound state (Fig 2B), with antibody binding inducing repositioning of the α-helix from outside the barrel to in-between the s2-s3 and s4-s5 loops. To facilitate the large α-helix the cavity between these loops enlarges via partial disassembly of the canonical fold's β-barrel. Some curvature inversion occurs within the strands of the barrel to facilitate this process, particularly for the s2 and s3 strands (Fig 2A/2B). In the apo state the tips of the s2-s3 and s4-s5 loops are in direct contact while in the VHH$_{2nsp23}$-bound state they are 30Å apart.

Interestingly, movement of the s1 strand in the VHH$_{2nsp23}$-bound state results in an alternate β-barrel being created as part of an anti-parallel nsp9 homodimer in which two copies of s3 self-associate (Fig 2B). This homodimer is distinct from the form of nsp9 with parallel-aligned α-helices that predominates *in vitro* [16] or the antiparallel form reported for nsp9$_{229E}$ [26]. Most β-structural elements of nsp9$_{COV19}$ remain topologically adjacent to each other when apo or VHH$_{2nsp23}$-bound.

## Potential nsp9 hinge points

Although the s4 and s5 structural strands directly mediate VHH$_{2nsp23}$-binding their conformation is less obviously altered. They remain adjacent in both conformations and binding of the $^{VHH}$CDR3 loop appears to impinge on the position of the α-helix's N-terminus (*e.g.* Lys-92, Gly-93). Other key residues act as hinge points about which the larger-scale rearrangements occur. To get a clearer understanding of potential points of flexibility within the fold a Kleywegt plot was used to compare both states of nsp9 (Fig 2C). This highlighted how few changes are actually required to switch between the significantly different folding states. The comparative Ramachandran distances between each state were then mapped onto the structure of nsp9$_{COV19}$ (Fig 2D), highlighting a number of interesting points:

1. although there is some fraying in the first turn of αC its largescale movement is largely facilitated by the preceding backbone shifts in residues Lys-92/Gly-93 contacted directly by VHH$_{2nsp23}$;

2. strands s6 and s7 are basically unchanged except for two adjacent hinges midway along their length;

3. In the apo-state Leu-42 resides at a kink-point that curves s3 away from αC. In the VHH$_{2nsp23}$-bound state it forms a standard straight β-strand. In both states the sidechain of Leu-42 provides hydrophobic backing to the helix but transitions from one side of the strand to the other to do so.

4. Many of the residues within s2 alter, this strand undergoes perhaps the most significant topological movements.

5. Small residues at the tips of loop regions display flexibility (*e.g.* Gly-38/61) but result in minimal real space differences.

## Helical-interacting residues

During viral 5'-cap synthesis the N-terminus of nsp9$_{COV19}$ inserts into the catalytic NiRAN domain of nsp12 [9, 10]. Nsp12 engagement is mediated through two features of the apo-nsp9 fold: the GxxxG interaction-motif within the fold's C-terminal α-helix engages an external cleft of the NiRAN domain and the N-terminus inserts into the active site (Fig 3A) [9]. The

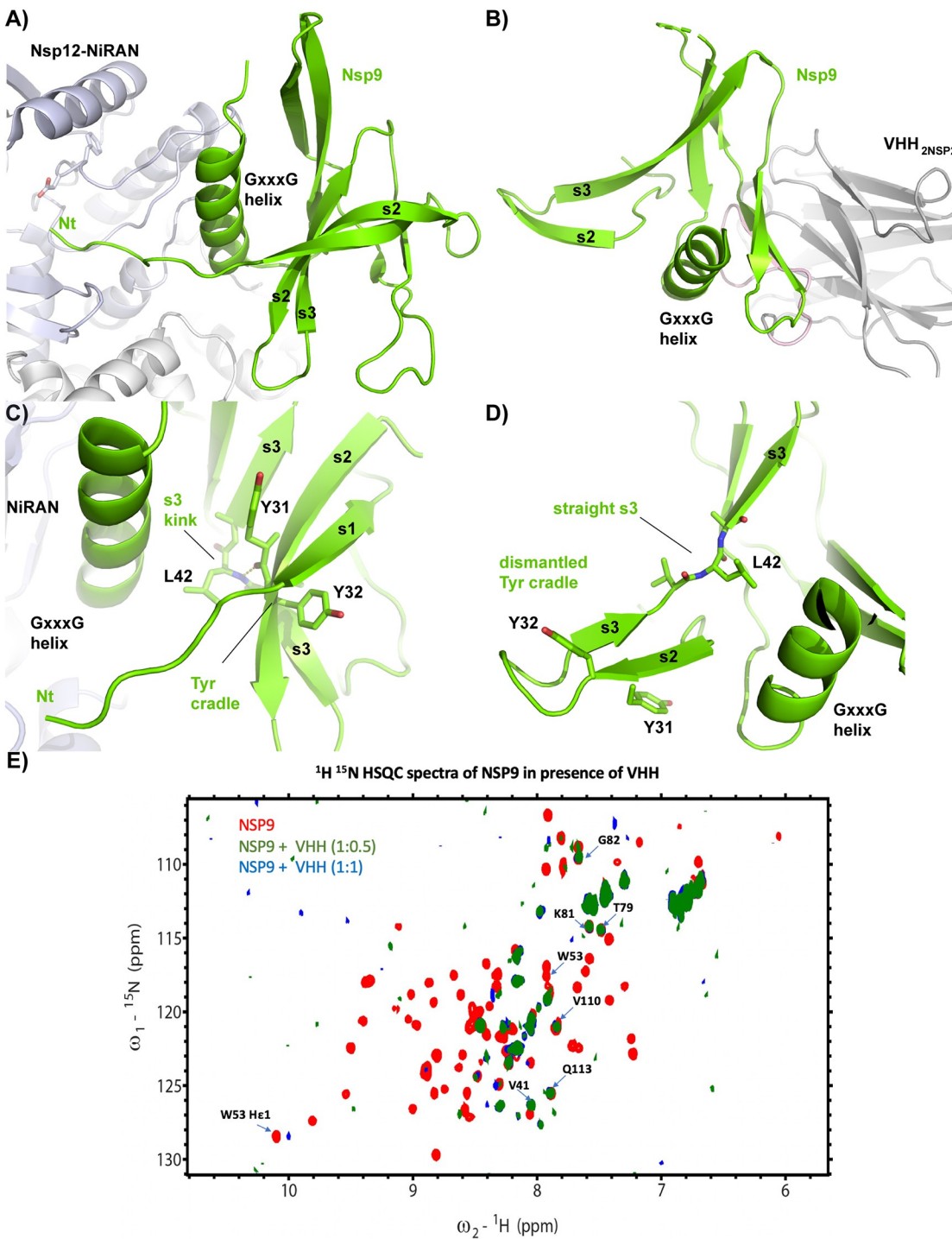

**Fig 3. Engagement of nsp9cov19 with the nsp12 NiRAN domain and VHH2nsp23. A)** Cartoon representation nsp9COV19 when engaged with the nsp12 NiRAN domain (10). **B)** Aligned nsp9COV19 structure when VHH2nsp23-bound demonstrating distortion of the NiRAN-engaging structural elements. **C)** Close-up view of the region surrounding the Leu-42 hinge in the apo state and **D)** VHH2nsp23-bound state. Residues discussed in text are labelled. **E)** Chemical perturbation of ${}^1$H${}^{15}$N HSQC spectra of labelled nsp9COV19 upon titration of VHH2nsp23.

new conformation of the s2-s3 loop and repositioned C-terminal helix appear to distort NiRAN-engaging elements of the fold in the VHH$_{2nsp23}$-bound state (Fig 3B).

In the NiRAN engaged state of nsp9 the protein's N-terminal residues project out from the body of the fold into the catalytic pseudokinase site [9]. Leu-42 acts as a kink within s3 (Fig 3C) that both interacts with the NiRAN associating α-helix and underlies an aromatic V-shaped tyrosine cradle that helps guide the protein's N-terminus towards the active site (Fig 3C). In the VHH$_{2nsp23}$-bound state s3 straightens out dismantling this cradle and its ability to guide the N-terminus which is largely disordered in the structure (Fig 3D, S1 Fig).

## In solution analysis of VHH-binding

The co-mixture of VHH$_{2Nsp23}$ and Nsp9$_{COV19}$ forms a higher molecular weight species in solution following co-complexation (S2 Fig). Previously we [18], and others [27, 28], had defined the backbone chemical shift assignments for nsp9$_{COV19}$. Due to the non-physiological nature of our crystallisation condition we sought to assess whether the co-complex we observe in crystals also occurs in solution via 1H-15N heteronuclear single quantum coherence (HSQC) nuclear magnetic resonance. The presence of VHH$_{2nsp23}$ resulted in widespread broadening of $^{15}$N-labelled nsp9$_{COV19}$ HSQC peaks, including that of Trp-53 (Fig 3E). This could be attributed to the increased size of the VHH$_{2nsp23}$:nsp9$_{COV19}$ complex. These results are broadly consistent with previously reported titration experiments and confirm a central role of Trp-53 within the epitope [25].

## Conclusion

A number of small molecule compounds with affinity for nsp9$_{COV19}$ have the potential to inhibit NiRAN engagement and hence viral RNAylation and capping [18, 19]. Small molecules investigated to date are only in preliminary phases of being developed and do not yet present truly nsp9-specific reagents. The larger interface utilised by antibodies for antigen-binding facilitates formation of highly specific high affinity interactions and have been investigated as potential COVID-19 therapeutics [29–31]. Most effort on such antibody-therapeutics have been directed towards virion proteins such as the SARS-CoV-2 Spike, but those binding to other viral proteins have nevertheless shown promise [25].

A cohort of nsp9$_{COV19}$-specific llama antibodies had previously been developed [25]. We further define the binding epitope of one of these, VHH$_{2nsp23}$, by solving its structure in complex with nsp9$_{COV19}$. The binding epitope was observed to be extensive utilising both the [VHH]CDR2 and [VHH]CDR3 loops. The antibodies epitope is directed towards residues within the s4 and s5 strands of nsp9, being particularly focused upon Trp-53 which had previously been suggested by NMR chemical peak perturbation assays [25]. Surprisingly antibody binding to nsp9$_{COV19}$ also resulted in large scale reorientation of secondary structural elements within the small coronaviral-unique fold. The C-terminal protein-interacting α-helix of nsp9$_{COV19}$ undergoes a large real-space shift from outside the fold's mini β-barrel to becoming encased by its central elements. The new VHH$_{2nsp23}$-bound architecture of nsp9$_{COV19}$ represents a novel fold.

This new architecture is significant as the C-terminal α-helix of nsp9 contains a conserved GxxxG protein-interaction motif once considered important for homodimer formation [16, 20, 26] but which is now known to provide a major portion of the interface for nsp12 NiRAN engagement [9]. Thus, the GxxxG-motif of nsp9 is indirectly responsible for mediating RNAylation [21]. In the VHH-bound state this α-helix become cryptically-encased. Thus, VHH$_{2nsp23}$ appears to stabilise a form of nsp9$_{COV19}$ unproductive for viral 5'-cap formation. This could be

therapeutically advantageous if small molecule reagents could be found that also induce this transition.

While coronaviral nsp9 has a relatively small fold it is also known to be somewhat flexible. In prior crystal structures its N-terminus was often poorly resolved and its manner of self-association *in vitro* can occur with angular variation in helix-to-helix contacts [16, 32]. Moreover, multiple configurations of the s4-s5 loop have been observed in high-resolution structures when in complex with RNA base-like compounds [18]. However, the changes described following VHH$_{2nsp23}$-binding involve a far more profound degree of structural reorganisation.

In the unliganded state the s2-s3 and s4-s5 loops project from one end of the fold's mini β-barrel forming a cavity that had previously been proposed as a potential ssRNA-binding site [16]. In our VHH$_{2nsp23}$-bound state the tips of these loops have opened extensively to expose the hydrophobic residues below. The fold's GxxxG α-helix swivels about its N-terminus to reside in this newly enlarged cavity. To accommodate this movement the s2-s3 loop undergoes the most significant changes, of particular note is a sharp kink in the apo state's s3 strand at Leu-42 halfway along its length which straightens upon VHH$_{2nsp23}$ binding resulting in dismantling of the aromatic cradle that helps position nsp9's N-terminus as it enters the NiRAN active site.

Is this new VHH$_{2nsp23}$-bound state of nsp9 physiologically relevant or is merely induced by the extensive set of VHH interactions energetically forcing the new state? Although the latter scenario seems most plausible the role of nsp9 during coronaviral replication is still unclear. In the model of Park *et al* two cycles of enzymatic activity are proposed from the NiRAN domain during GpppA-RNA cap synthesis [21]. We note that our VHH$_{2nsp23}$-bound structure appears to allow more freedom for the protein's N-terminus following loss of the Tyr-cradle. Our work highlights a number of structural points within the nsp9 fold that may be capable of acting as hinge points. Moreover, we demonstrate that VHH-binding, which occurs at a site distal to the NiRAN domain-interaction site, may nonetheless result in structural changes capable of disrupting a protein-interface. The potential for alternate forms of nsp9, as well as issues over its oligomerisation state, should to some extent be kept in mind when assessing *in vivo* mutational experiments.

## Experimental procedures

### Protein expression and purification

Recombinant nsp9 with an enterokinase-cleavable His-tag that leaves a scar-free protein was produced as described previously [32].

### 2NSP23 production, expression and purification

The sequence for VHH$_{2nsp23}$ and related nanobody cohort [25] was made synthetically and cloned into a pET-30 *E. coli* expression vector. VHH$_{2nsp23}$ was expressed as an inclusion body by growing transformed *E. coli* BL21 in Luria-Bertani broth to an Abs$_{600}$ of 1.0 then inducing it for 6 hours at 37˚C and harvesting it in buffer A (50 mM Tris pH 8, 150 mM NaCl). Inclusion bodies (IB) were isolated by sonication then centrifuged at 15000g and washed repeatedly in the presence of buffer supplemented with 1% v/v Triton X-100 and 1 mM DTT. IBs were resuspended in 6 M Guanadinium.HCl then refolded for 24hrs in 50 mM Tris pH 8.0, 100 mM NaCl, 0.5 M Arginine, 1 M Urea, 1 mM GSH, 0.2 mM GSSG before being dialysed against buffer A and purified on DEAE affinity resin and gel-filtration chromatography.

## Complex crystallisation, data collection and structure determination

Nsp9$_{COV19}$ with the His-tag removed was complexed with VHH$_{2nsp23}$ then purified on a Superdex S75 10/300 gel filtration column before concentrating to ~15 mg/mL and crystallised via hanging drop vapour diffusion over a reservoir consisting of 1.8 M ammonium sulfate, 0.1 M MES pH 6.0. Crystals were transferred into cryoprotectant containing an additional 24% v/v glycerol then frozen in liquid nitrogen. All diffraction data were collected at the Australian synchrotrons MX2 beamline [33]. Data was processed using the program XDS [34], scaled and merged with programs from the CCP4 suite [35]. Diffraction data was processed in P6. Initial phases were obtained using the molecular replacement program Phaser [36] with a trimmed VHH model. Subsequent rounds of manual building and refinement were performed in the programs COOT [37] and Phenix [38].

## HSQC VHH measurements

$^{1}$H-$^{15}$N HSQC spectra of apo labelled nsp9$_{COV19}$ was recorded as reported previously [18]. Spectra were then recollected after unlabelled VHH$_{2nsp23}$ was titrated into the sample at a 2:1 ratio and a 1:1 ratio of antibody to nsp9.

## Ramachandran distance calculations

The output of DSSP [39] was used to calculate Phi / Psi angles for the both states. Ramachandran distance mapped to shortest possible quadrant was then calculated using:

$$dist = \sqrt{\left(\phi_A - \phi_{VHH}\right)^2 - \left(\varphi_A - \varphi_{VHH}\right)^2}$$

## Supporting information

**S1 Table. Data collection and refinement statistics.**
(DOCX)

**S1 Fig. Overlay of crystallographic B-factor on the Nsp9$_{COV19}$:VHH$_{2Nsp23}$ complex.** The final model was coloured with a blue-green-red gradient according to the temperature factors of the C$_\alpha$ atoms.
(PNG)

**S2 Fig. VHH Co-complexation.** *Left*: Superdex S75 10/300 gel-filtration chromatograms following injection of ~5mg of VHH$_{2Nsp23}$, Nsp9$_{COV19}$ or a pre-incubated mixture. *Right*: Native gel electrophoresis of ~100μg of VHH$_{2Nsp23}$, Nsp9$_{COV19}$ or the complex together.
(PNG)

## Acknowledgments

This research was undertaken in part using the MX beamlines at the Australian Synchrotron, part of ANSTO, and made use of the Australian Cancer Research Foundation (ACRF) detector. Additionally, we thank Dr. Geoffrey Kong of the Monash Molecular Crystallisation Facility for his assistance with crystallographic screening and optimization. We thank A. Riboldi-Tunnicliffe and R. Williamson for assistance with data collection. JR is supported by an Australian Research Council Laureate Fellowship.

## Author Contributions

**Conceptualization:** Dene R. Littler.

**Data curation:** Indu R. Chandrashekaran.

**Formal analysis:** Indu R. Chandrashekaran, Dene R. Littler.

**Investigation:** Yue Pan, Luke Tennant.

**Writing – original draft:** Dene R. Littler.

**Writing – review & editing:** Jamie Rossjohn.

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
