## [Decision Letter · Decision Letter 0]

25 Oct 2022

PONE-D-22-24483Inside-out: antibody-binding reveals potential folding hinge-points within the SARS-CoV-2 replication co-factor Nsp9PLOS ONE

Dear Dr. Littler,

Thank you for submitting your manuscript to PLOS ONE. I also apologize for my tardiness in getting your manuscript reviewed. After careful consideration, we feel that it has merit but does not fully meet PLOS ONE’s publication criteria as it currently stands. Therefore, we invite you to submit a revised version of the manuscript that addresses the points raised during the review process.

       My recommendation for major revision is based on the comments from one of the reviewers. However, I believe these revisions are feasible.

        Please submit your revised manuscript by Dec 09 2022 11:59PM. If you will need more time than this to complete your revisions, please reply to this message or contact the journal office at plosone@plos.org. Please include the following items when submitting your revised manuscript:A rebuttal letter that responds to each point raised by the academic editor and reviewer(s). You should upload this letter as a separate file labeled 'Response to Reviewers'.A marked-up copy of your manuscript that highlights changes made to the original version. You should upload this as a separate file labeled 'Revised Manuscript with Track Changes'.An unmarked version of your revised paper without tracked changes. You should upload this as a separate file labeled 'Manuscript'.

We look forward to receiving your revised manuscript.

Kind regards,

Michael Massiah

Academic Editor

PLOS ONE

Journal Requirements:

3. Thank you for stating in your financial disclosure:  

"The work is funded by internal Departmental support." 

PLOS ONE requires you to include in your manuscript further information about the funder so that any relevant competing interests can be assessed. Please respond to the following questions:

a) Please state whether any of the research costs or authors' salaries were funded, in whole or in part, by a tobacco company (our policy on tobacco funding is at http://journals.plos.org/plosone/s/disclosure-of-funding-sources)  

b) Please state whether the donor has any competing interests in relation to this work (see http://journals.plos.org/plosone/s/competing-interests) . 

c) Please state whether the identity of the donor might be considered relevant to editors or reviewers’ assessment of the validity of the work.

d) If the donors have no perceived or actual competing interests, please state: “The authors are not aware of any competing interests”. 

This information should be included in your cover letter. We will amend your financial disclosure and competing interests on your behalf.

"There are no competing interests."

5.In your Data Availability statement, you have not specified where the minimal data set underlying the results described in your manuscript can be found. PLOS defines a study's minimal data set as the underlying data used to reach the conclusions drawn in the manuscript and any additional data required to replicate the reported study findings in their entirety. All PLOS journals require that the minimal data set be made fully available. For more information about our data policy, please see http://journals.plos.org/plosone/s/data-availability.

6. Please amend your authorship list in your manuscript file to include author "Yue Pan".

8. Please upload a copy of Supporting Information Table 1 which you refer to in your text on page 3.

Reviewers' comments:

Reviewer's Responses to Questions

**Comments to the Author**

1. Is the manuscript technically sound, and do the data support the conclusions?

Reviewer #1: Yes

Reviewer #2: Yes

Reviewer #3: Yes

2. Has the statistical analysis been performed appropriately and rigorously? 

Reviewer #1: N/A

Reviewer #2: N/A

Reviewer #3: N/A

3. Have the authors made all data underlying the findings in their manuscript fully available?

Reviewer #1: Yes

Reviewer #2: Yes

Reviewer #3: Yes

4. Is the manuscript presented in an intelligible fashion and written in standard English?

Reviewer #1: Yes

Reviewer #2: Yes

Reviewer #3: Yes

5. Review Comments to the Author

Reviewer #1: This manuscript reports a structure of SARS-CoV-2 nsp9 bound to nanobody VHH2NSP23. Identification of the binding epitope residues will aid future drug development. Changes in the GxxxG helix, which is responsible for Nsp9-NiRAN interactions, moving to the cavity between the s2-s3 and s4-s5 loops, are particularly noteworthy. Understanding the Nsp9/NiRAN interactions is essential to solving the CoV-2 capping mechanism. Although several models have been proposed, the alternative structure of nsp9 revealed in this study may open new possibilities for research. For example, are there native binding partners of nsp9 that can induce these conformational changes? If yes, will it change the capping pathway?

The manuscript is clearly written and well illustrated. I suggest a few minor changes

1) Page 2 line 4, change ‘Nsp12’ to ‘nsp12’. All viral nsps should be in lowercase throughout the manuscript.

2) Page 3 line 9, active NMPylation can be supported by Mg2+. Ref “NMPylation and de-NMPylation of SARS-CoV-2 nsp9 by the NiRAN domain, Nucleic Acids Research, https://doi.org/10.1093/nar/gkab677”. Line 26, I suggest deleting ‘compromising binding’ here since no data is supporting it in this manuscript. Lines 41-43 are very confusing, please rewrite the sentence. Maybe separate it into two sentences.

3) Page 4 lines 13 – 14, does this description correspond to Figure 1C?

4) Page 7 line 3, if the issue of ‘We note that crystallization conditions can be unusual’ was solved by HSQC, this sentence can be deleted. line 14, 6 hours. line 15, 50 mM, please check other places. Line 16, in the presence of buffer A? line 35, ‘1:0.5 ratio’? It’s better to specify the ratio of what to what.

5) ‘VHH2NSP23’ or ‘VHH’? Please choose one and stick to it.

6) Are ‘nsp9’ and ‘nsp9COV19’ different? The use of them is somewhat confusing.

7) Ref 21 has been published.

8) The names in ‘Author Contributions’ part don’t match the authors in the titles.

9) Figure 1 legend, buried surface area (BSA); what’s ‘germ’? where is the BSA of CDR1? And please change the color of CDR1 since green is occupied by nsp9. Use ‘kDa’ for the marker of the SDS-PAGE in (A). Not mandatory, but I suggest redoing the SDS-PAGE if possible. The nsp9 band can barely be seen. In (C), change CDR2 residues to blue.

10) In figure 2(A), not all of these colored key residues are labeled. (B) the ‘s6’ has a different font. (D) what’s the meaning of the color gradient? (E) what’s the meaning of the underlines?

Reviewer #2: Pone-D-22-24483

A paper describing the structure of SARS-CoV-2 nsp9 complexed with a nanobody. Nsp9 undergoes a rearrangement with the nanobody binding, disrupting a conserved Beta-sheet and creating new alpha-helices in an inside out pattern. The area changed by the nanobody is thought critical to nsp9 function in the initial viral-mRNA capping reaction, and the authors speculate that this interruption of nsp9 function could be therapeutic in nature. They also speculate that the novel structure could be a naturally occurring as a structural change that occurs in nsp9 during its functions during viral infection. A few comments to address on revision

1. The authors speculate that antibodies or nanobodies that have this function of binding nsp9 and inducing the change in structure could be therapeutic during viral infection. They mention in example, therapeutic antibodies that are against spike. I think these anti-nsp9 nanobodies or antibodies would have to delivered intracellularly and make it to the double membrane enclosed replication complex of SARS-CoV-2. I think the authors should point this out and give ideas for such intracellular delivery.

2. Line 32 on page 5, “guild” should probably be “guide”

Reviewer #3: Comments to authors:

In this manuscript, Pan et al. characterized the interactions between SARS-CoV-2 Nsp9 protein with an existing anti-Nsp9 nanobody using X-ray crystallography. The authors solved the crystal structure of the Nsp9-nanobody complex at 2.4 Å resolution. They confirmed that the binding interactions are centered around a Trp-53 within the Nsp9, consistent with the previous NMR-based structural studies of the same complex. They also found that binding the nanobody brings conformational changes in the Nsp9, hampering the Nsp9 binding with its natural substrate NiRAN domain of Nsp12. However, the manuscript has not explicitly discussed the mechanism of such conformational change, which may be a consequence of the nanobody binding.

The manuscript is well-written and presents an important advancement in the structural understanding of SARS-CoV-2 proteins and their complexes with antibodies, which may help develop alternative strategies for COVID-19 prevention and control. Broadly, conclusions are supported by the experiments, but several points are discussed poorly. For example, the authors said that “the new VHH-bound architecture of Nsp9COV19 represents a novel fold”, but it is unclear how and why it is a “novel” fold of a protein. I expect the authors consider the following points to revise the manuscript before it can be accepted for publication.

1) VHH should be defined when used first-time in the text. It is unclear until reading down to the results section.

2) Fig.1A: The labels on top of the bands make them difficult to see. I assumed these were the MW markers.

3) The crystallographic asymmetric unit contained two complexes. It is unclear whether these complexes tend to dimerize in solution. Some tests, such as SEC or native gel electrophoresis, will be helpful.

4) What is the affinity of Nsp9 binding with VHH? How does it differ from Nsp9 binding with the NiRAN domain? It may give a clearer perspective on the energetics of Nsp9 deformations caused by the VHH binding.

5) Two Nsp9 molecules contact each other heavily within the crystallographic asymmetric unit. Does Nsp9 dimerize in the solution? Does this contribute to the observed deformation of the Nsp9 upon VHH binding? It would be helpful to include this discussion.

6) The authors discussed that Nsp9 Trp-53 is the primary binding site for the VHH. However, they did not discuss how Trp53 mutation impacts this binding.

7) A figure with crystallographic B-factors will be helpful to see rigid and more dynamic regions in the Nsp9-VHH complex.

8) Crystallographic data collection and refinement statistics table should be presented (in the manuscript or the SI).

6. PLOS authors have the option to publish the peer review history of their article (what does this mean?). If published, this will include your full peer review and any attached files.

Reviewer #1: No

Reviewer #2: **Yes: **Wesley C Van Voorhis

Reviewer #3: No

---

## [Author Response · Author response to Decision Letter 0]

31 Jan 2023

Reviewer #1: This manuscript reports a structure of SARS-CoV-2 nsp9 bound to nanobody VHH2NSP23. Identification of the binding epitope residues will aid future drug development. Changes in the GxxxG helix, which is responsible for Nsp9-NiRAN interactions, moving to the cavity between the s2-s3 and s4-s5 loops, are particularly noteworthy. Understanding the Nsp9/NiRAN interactions is essential to solving the CoV-2 capping mechanism. Although several models have been proposed, the alternative structure of nsp9 revealed in this study may open new possibilities for research. For example, are there native binding partners of nsp9 that can induce these conformational changes? If yes, will it change the capping pathway?

The manuscript is clearly written and well illustrated. I suggest a few minor changes

1) Page 2 line 4, change ‘Nsp12’ to ‘nsp12’. All viral nsps should be in lowercase throughout the manuscript.

We thank reviewer1 for the comment, 125 replacements were made – excepting those where Nsp9 is at the beginning of a sentence.

2) Page 3 line 9, active NMPylation can be supported by Mg2+. Ref “NMPylation and de-NMPylation of SARS-CoV-2 nsp9 by the NiRAN domain, Nucleic Acids Research.

Agreed we have altered to the following and added reference to end of sentence 

“…Mn2+-dependent NMPylase…” � “…Mg2+ or Mn2+-dependent NMPylase…” 

Line 26, I suggest deleting ‘compromising binding’ here since no data is supporting it in this manuscript. 

Done

Lines 41-43 are very confusing, please rewrite the sentence. Maybe separate it into two sentences.

Changed Page 3, line 41 -> “Both components where present in our crystals when run on SDS-PAGE (Fig 1A) but our initial attempts to obtain the antibody-bound nsp9COV19 structure was unsuccessful.”

3) Page 4 lines 13 – 14, does this description correspond to Figure 1C?

Correct, added a panel callout �“…Gly-93 respectively (Fig 1C)”

4) Page 7 line 3, if the issue of ‘We note that crystallization conditions can be unusual’ was solved by HSQC, this sentence can be deleted. 

Deleted as recommended.

line 14, 6 hours. line 15, 50 mM, please check other places. Line 16, in the presence of buffer A? line 35, ‘1:0.5 ratio’? It’s better to specify the ratio of what to what.

“6hours” -> “6 hours”

14 similar replacements in methods

To ratio added “of antibody to nsp9”

5) ‘VHH2NSP23’ or ‘VHH’? Please choose one and stick to it.

14 substitutions to standardise this, mainly converting VHH-bound to VHH2nsp23-bound

6) Are ‘nsp9’ and ‘nsp9COV19’ different? The use of them is somewhat confusing.

Added “…in complex with the SARS-CoV-2 nsp9 (we term nsp9COV19)

I have used this to distinguish between when we are writing about all coronaviral nsp9 and a specific viral variant. We feel this is important to distinguish as nsp9SARS is very similar to nsp9COV19 while nsp9MERS has considerable differences and nsp9HKU is almost unrecognisable. They presumably all control 5’ capping but ideally any NiRAN/nsp9-directed inhibitors should target as wide a range as possible. Viral variant variability also represents potential escape mutants. 

7) Ref 21 has been published.

Excellent, updated accordingly. We also needed to change page 6, line 44 to Park et al. due to the altered first author of the paper.

8) The names in ‘Author Contributions’ part don’t match the authors in the titles.

Corrected

9) Figure 1 legend, buried surface area (BSA); what’s ‘germ’? where is the BSA of CDR1? 

Apologies, changed to the following: 

“The doughnut chart insert shows the respective contributions to the buried surface area (BSA) for the antibodies CDR-loop and germline residues.”

And please change the color of CDR1 since green is occupied by nsp9. 

Good idea, altered to orange.

Use ‘kDa’ for the marker of the SDS-PAGE in (A). 

Added MW (kDa) to Fig 1A

Not mandatory, but I suggest redoing the SDS-PAGE if possible. The nsp9 band can barely be seen. 

Sorry, this is the SDS-PAGE of the crystal from which the dataset was collected, i.e. it is there to try and show what was in the crystal. I agree it’s an awful gel but was done on a washed 100um3 crystal sample.

In (C), change CDR2 residues to blue.

Replaced with the following

10) In figure 2(A), not all of these colored key residues are labeled. 

– Now label Y87

(B) the ‘s6’ has a different font. 

– fixed in panel B

(D) what’s the meaning of the color gradient? 

Added to the caption:

“The Ramachandran distances between the two states are mapped onto the canonical nsp9cov19 structure using a white->pink->red gradient where the upper end of the scale is theoretical maximum”

(E) what’s the meaning of the underlines?

Removed underlines – they were highlighting the regions undergoing structural change but are somewhat redundant with panel D.

Reviewer #2: Pone-D-22-24483

A paper describing the structure of SARS-CoV-2 nsp9 complexed with a nanobody. Nsp9 undergoes a rearrangement with the nanobody binding, disrupting a conserved Beta-sheet and creating new alpha-helices in an inside out pattern. The area changed by the nanobody is thought critical to nsp9 function in the initial viral-mRNA capping reaction, and the authors speculate that this interruption of nsp9 function could be therapeutic in nature. They also speculate that the novel structure could be a naturally occurring as a structural change that occurs in nsp9 during its functions during viral infection. A few comments to address on revision

1. The authors speculate that antibodies or nanobodies that have this function of binding nsp9 and inducing the change in structure could be therapeutic during viral infection. They mention in example, therapeutic antibodies that are against spike. I think these anti-nsp9 nanobodies or antibodies would have to delivered intracellularly and make it to the double membrane enclosed replication complex of SARS-CoV-2. I think the authors should point this out and give ideas for such intracellular delivery.

We agree with this point whole-heartedly! Intracellular delivery of VHHs in live cells is possible but not the preferred option in our opinion. We note the referenced Espositio’s paper had suggested it is a viable possibility though. We see the structural transition induced by the VHH to be informative in itself due to its stabilisation of this new semi-stable secondary form of nsp9. Now we know this nsp9 form can be stabilised we are in the process of trying to induce its formation with cell-permeable cyclic peptides based on the VHHs CDR3 loop. The text has been altered to make this clearer without being too specific about our current work via the following changes:

Page 6, line 4 -> 

“Most effort on such antibody-therapeutics have been directed towards virion proteins such as the SARS-CoV-2 Spike, but those binding to other viral proteins have nevertheless shown promise (25).”

Page 6, line 22 ->

“This could be therapeutically advantageous if small molecule reagents could be found that also induce this transition.”

2. Line 32 on page 5, “guild” should probably be “guide”

Well spotted, typo has been fixed.

Reviewer #3: Comments to authors:

In this manuscript, Pan et al. characterized the interactions between SARS-CoV-2 Nsp9 protein with an existing anti-Nsp9 nanobody using X-ray crystallography. The authors solved the crystal structure of the Nsp9-nanobody complex at 2.4 Å resolution. They confirmed that the binding interactions are centered around a Trp-53 within the Nsp9, consistent with the previous NMR-based structural studies of the same complex. They also found that binding the nanobody brings conformational changes in the Nsp9, hampering the Nsp9 binding with its natural substrate NiRAN domain of Nsp12. However, the manuscript has not explicitly discussed the mechanism of such conformational change, which may be a consequence of the nanobody binding.

The manuscript is well-written and presents an important advancement in the structural understanding of SARS-CoV-2 proteins and their complexes with antibodies, which may help develop alternative strategies for COVID-19 prevention and control. Broadly, conclusions are supported by the experiments, but several points are discussed poorly. For example, the authors said that “the new VHH-bound architecture of Nsp9COV19 represents a novel fold”, but it is unclear how and why it is a “novel” fold of a protein. I expect the authors consider the following points to revise the manuscript before it can be accepted for publication.

1) VHH should be defined when used first-time in the text. It is unclear until reading down to the results section.

Added to Page 3 line 13 ->

“Nanobodies are isolated variable heavy domains from camelid immunoglobulins, also termed VHHs.” 

2) Fig.1A: The labels on top of the bands make them difficult to see. I assumed these were the MW markers.

Well spotted, this has been fixed.

3) The crystallographic asymmetric unit contained two complexes. It is unclear whether these complexes tend to dimerize in solution. Some tests, such as SEC or native gel electrophoresis, will be helpful.

Excellent suggestion, we have included the following as Supporting figure 2 which details the shift towards higher MW species.

Supporting Figure 2 – VHH Co-complexation. 

Left: Superdex S75 10/300 gel-filtration chromatograms following injection of ~5mg of VHH2Nsp23, Nsp9COV19 or a pre-incubated mixture. Right: Native gel electrophoresis of ~100�g of VHH2Nsp23, Nsp9COV19 or the complex together.

Added: Page 5, line 37 -> “The co-mixture of VHH2Nsp23 and Nsp9COV19 forms a higher molecular weight species in solution following co-complexation (Supp. Fig. 2).”

4) What is the affinity of Nsp9 binding with VHH? How does it differ from Nsp9 binding with the NiRAN domain? It may give a clearer perspective on the energetics of Nsp9 deformations caused by the VHH binding.

This is an excellent question but one I cannot answer at present. The affinity of Nsp9 for the NiRAN domain is hard to measure as this is a multi-component measurement. That is, the native affinity is for the large multi-subunit complex consisting of 5’-RNA:Nsp12:Nsp8 x 2:Nsp7:Nsp13 x 2. Due to the nature of RNA loading within this complex I am not aware of anyone making enough of this 300kDa nucleo-protein structure to consider measuring an affinity. Published EM single-particle reconstructions (Pang et al 2022) involve reconstructions of Nsp9-bound forms from tiny amounts of sample. At present the affinity between Nsp9 and RNA/NiRAN domain is probably thought to be low, but the difficultly in ensuring RNA is loaded correctly means I wouldn’t be sure enough to put this in a manuscript.

A transient and conditional association of Nsp9 could make biological sense. 5’-capping must be a tightly regulated interaction i.e. a misplaced cap would “waste” an entire viral genome and potentially lead to an immune-visible signature.

5) Two Nsp9 molecules contact each other heavily within the crystallographic asymmetric unit. Does Nsp9 dimerize in the solution? Does this contribute to the observed deformation of the Nsp9 upon VHH binding? It would be helpful to include this discussion.

In solution in vitro Nsp9 adopts a well-characterised homodimeric form first characterised by Sutton et al. 2004 for SARS-CoV and which we and others have shown is also adopted for SARS-CoV-2 Nsp9. This homodimeric form has similar monomer structure to the NiRAN-engaged monomeric form of Nsp9 observed in cryo-EM structures. 

In this manuscript the VHH-bound homodimeric form of Nsp9 is distinct from the previously published homodimer as shown in the movie. At present it is not currently clear to what extent homodimeric forms of Nsp9 are physiological. Nsp9 requires some coaxing to associate with the NiRAN domain in vitro but as described above contributions from the viral RNA enhance binding in single particle cryo-EM analysis. Viral RNA will be more plentiful and stable in an infected cell, distal sections of the viral genome may also enhance Nsp9 binding and we may be missing these at present. 

6) The authors discussed that Nsp9 Trp-53 is the primary binding site for the VHH. However, they did not discuss how Trp53 mutation impacts this binding.

Tryptophan has a characteristic NMR signature making their cross-peaks easy to identify. Chemical shifts on the Trp-53 peak had previously been used to define it as contributing to the VHH23’s epitope (Esposito et al. 2021). This is why we talk about it being central to the epitope and further characterise its interactions. But I the word centrality here to refer to Trp-53 location near the middle of the binding site not to mean it was the sole-contributor to affinity. Other residues mentioned such as Glu-68 and Tyr-66 and Ile-65 make as many interactions upon binding. My assumption would be that binding-affinity is derived from the sum of such interactions. i.e. a simple ala mutation at each site would reduce binding affinity rather than abrogate it. We note that no variations at this site have thus far been observed for SARS-CoV-2 variants and this residue is a Trp in SARS-CoV but not 229E.

To make this point clearer I have changed:

Page 4, line 20 – “centrality” changed to “contribution”

7) A figure with crystallographic B-factors will be helpful to see rigid and more s

Supporting Figure 1 - Overlay of crystallographic B-factor on the Nsp9COV19:VHH2Nsp23 complex. The final model was coloured with a blue-green-red gradient according to the temperature factors of the C� atoms.

Page 5, line 34 (Supp. Fig 1.)

8) Crystallographic data collection and refinement statistics table should be presented (in the manuscript or the SI).

This is now included in Supp. Table 1 and referenced on page 3, line 37

---

## [Decision Letter · Decision Letter 1]

20 Feb 2023

PONE-D-22-24483R1Inside-out: antibody-binding reveals potential folding hinge-points within the SARS-CoV-2 replication co-factor Nsp9PLOS ONE

Dear Dr. Littler,

Thank you for submitting your manuscript to PLOS ONE. After careful consideration, we feel that it has merit but does not fully meet PLOS ONE’s publication criteria as it currently stands. Therefore, we invite you to submit a revised version of the manuscript that addresses the points raised during the review process.

  Please note that despite the reviewer agreeing that the revise manuscript is acceptable, there are concerns that the reviewer feels that the authors did not adequately address or address suitable.

We look forward to receiving your revised manuscript.

Kind regards,

Michael Massiah

Academic Editor

PLOS ONE

Journal Requirements:

Reviewers' comments:

Reviewer's Responses to Questions

**Comments to the Author**

1. If the authors have adequately addressed your comments raised in a previous round of review and you feel that this manuscript is now acceptable for publication, you may indicate that here to bypass the “Comments to the Author” section, enter your conflict of interest statement in the “Confidential to Editor” section, and submit your "Accept" recommendation.

Reviewer #3: All comments have been addressed

2. Is the manuscript technically sound, and do the data support the conclusions?

Reviewer #3: Yes

3. Has the statistical analysis been performed appropriately and rigorously? 

Reviewer #3: N/A

4. Have the authors made all data underlying the findings in their manuscript fully available?

Reviewer #3: Yes

5. Is the manuscript presented in an intelligible fashion and written in standard English?

Reviewer #3: Yes

6. Review Comments to the Author

Reviewer #3: Authors have addressed most of the issues raised in previous version. However, some of the things authors said "done" in the response letter are not reflected in the revision. For example, the Fig. 1A still contains labels on top of the bands.

7. PLOS authors have the option to publish the peer review history of their article (what does this mean?). If published, this will include your full peer review and any attached files.

Reviewer #3: No

---

## [Author Response · Author response to Decision Letter 1]

26 Feb 2023

Reviewer #3: Authors have addressed most of the issues raised in previous version. 

However, some of the things authors said "done" in the response letter are not reflected in the revision. For example, the Fig. 1A still contains labels on top of the bands.

Apologies, version error.

Fig. 1A changed to ...

---

## [Editor Report · Decision Letter 2]

6 Mar 2023

Inside-out: antibody-binding reveals potential folding hinge-points within the SARS-CoV-2 replication co-factor Nsp9

PONE-D-22-24483R2

Dear Dr. Littler,

We’re pleased to inform you that your manuscript has been judged scientifically suitable for publication and will be formally accepted for publication once it meets all outstanding technical requirements.

Kind regards,

Michael Massiah

Academic Editor

PLOS ONE
---

## [Editor Report · Acceptance letter]

30 Mar 2023

PONE-D-22-24483R2 

Inside-out: antibody-binding reveals potential folding hinge-points within the SARS-CoV-2 replication co-factor Nsp9 

Dear Dr. Littler:

I'm pleased to inform you that your manuscript has been deemed suitable for publication in PLOS ONE. Congratulations! Your manuscript is now with our production department. 

Kind regards, 

on behalf of

Dr. Michael Massiah 

Academic Editor

PLOS ONE